# Fine-Grained Side Information Guided Dual-Prompts for Zero-Shot Skeleton Action Recognition

## ABSTRACT

Skeleton-based zero-shot action recognition aims to recognize unknown human actions based on the learned priors of the known skeleton-based actions and a semantic descriptor space shared by both known and unknown categories. However, previous works mostly focus on establishing the bridges between the known skeleton representation space and semantic descriptions space at the coarse-grained level for recognizing unknown action categories, ignoring the fine-grained alignment of these two spaces, resulting in suboptimal performance in distinguishing high-similarity action categories. To address these challenges, we propose a novel method via **S**ide information and dual-promp**T**s learning for skeleton-based zero-shot **A**ction **R**ecognition (STAR) at the fine-grained level. Specifically, 1) we decompose the skeleton into several parts based on its topology structure and introduce the side information concerning multi-part descriptions of human body movements for alignment between the skeleton and the semantic space at the fine-grained level; 2) we design the visual-attribute and semantic-part prompts to improve the intra-class compactness within the skeleton space and inter-class separability within the semantic space, respectively, to distinguish the high-similarity actions. Extensive experiments show that our method achieves state-of-the-art performance in ZSL and GZSL settings on NTU RGB+D, NTU RGB+D 120, and PKU-MMD datasets. The code will be available in the future.

## CCS CONCEPTS

• **Computing methodologies → Activity recognition and understanding**.

## KEYWORDS

Zero-Shot Learning, Skeleton-based Action Recognition, Side Information, Prompt Learning

## 1 INTRODUCTION

Many researchers pay attention to the action recognition community because of its wide range of applications, including intelligent monitors, sports analysis, anomaly action recognition [24], etc. Compared to the RGB-D modalities, human skeleton data (joint coordinate) has excellent robustness to the light intensity, background noise, and view variations. Meanwhile, it is easy to obtain skeleton pose data with the maturity of depth sensors like Kinect

*MM '24, October 28-November 1, 2024, Melbourne, Australia*
© 2024 Copyright held by the owner/author(s). Publication rights licensed to ACM.
ACM ISBN 978-1-4503-XXXX-X/18/06
https://doi.org/XXXXXXX.XXXXXXX

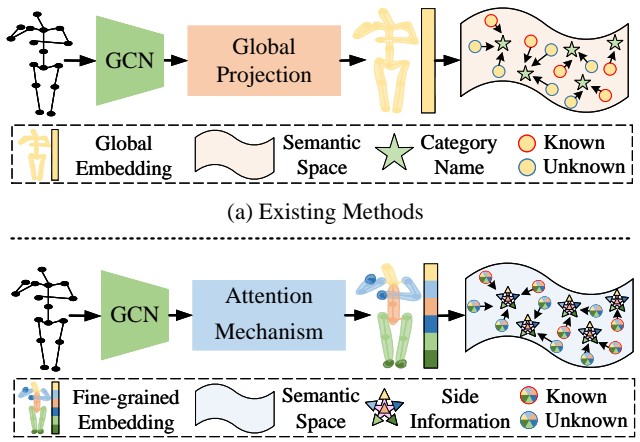

(a) Existing Methods

(b) Our STAR

Figure 1: Methods comparison. (a) Existing skeleton-based zero-shot action recognition methods project the global embedding of skeleton sequences into semantic space for alignment with category names, neglecting the potential correlation at the fine-grained level; (b) Our STAR decomposes the human skeleton into several regions based on its topology structure and introducing the extra side information of part motion descriptions for alignment at the fine-grained level, enabling significant capacities of transferring knowledge from known to unknown categories.

[33] and the development of pose estimation methods [34]. To this end, skeleton-based action recognition becomes a hot topic.

Existing skeleton-based action recognition studies focus on supervised or self-supervised learning to explore the spatial-temporal characteristics of the actions, which require training data to cover all the action categories that need to be recognized. However, these methods have some limitations, as follows. From the perspective of generality, they work well on recognizing the categories of actions that have appeared in the training set (known) but fail to recognize new categories outside of them (unknown), unlike the cognitive process of humans to recognize new categories based on existing knowledge. From another perspective of the costs, collecting and annotating endless action categories in the real world is unrealistic and expensive, especially for anomaly actions. Thus, zero-shot learning is employed in [9, 12, 13, 35] to address the challenges concerning recognizing unknown action categories without having access to the samples of unknown categories during training.

In practice, these works [9, 12, 13, 35] can be further divided into zero-shot learning (ZSL) and generalized zero-shot learning (GZSL) settings. The former only needs to classify unknown action categories during inference, while the latter also aims to classify actions of both known and unknown categories [21]. The core

of these methods is to align the known skeleton representations with the corresponding semantic embeddings (e.g., category names) by projecting them into the common space [12] or other learned metrics [8]. Then, the relationship of semantic embeddings between known and unknown categories from the pre-trained language models [18, 22, 23] shared in the same space can be utilized as the bridge to transfer knowledge from the known to the unknown. In this way, the learned model can recognize the skeleton action of the unknown categories according to their semantic information.

While these methods perform well in recognizing the skeleton actions of unknown categories, there are still several challenges. (1) These methods [9, 12, 13, 35] embed the skeleton sequences into the global representation for alignment with the corresponding coarse-grained semantic embeddings (e.g., category name), ignoring the potential correlation between fine-grained skeleton parts and semantics, as shown in Fig. 1(a). (2) The pre-extracted skeleton representations and semantic embeddings have low intra-class compactness and inter-class separability in their respective space, so it is challenging to distinguish the actions with high similarities. (3) In the GZSL, the previous studies [9, 13] designed the gating module through the probability distribution over known categories to classify unknown categories, which is unreasonable. These problems limit the model's generalization power, resulting in challenges to transferring knowledge from the known to unknown categories.

To address the above challenges, we propose a novel method via **S**ide information and dual-Promp**T**s learning for zero-shot skeleton **A**ction **R**ecognition (STAR) as shown in Fig. 1(b). Specifically, we decompose the skeleton sequence into several parts based on the human spatial topological structure. Then, we utilize the GPT-3.5 [1] as the expert knowledge base to generate the motion descriptions of each skeleton part as the side information, enriching the part spatial-temporal information of category names. However, these skeleton parts and corresponding side information exhibit homogeneity among high-similarity actions. Thus, we introduce the learnable visual-attribute prompts and the cross-attention mechanism in each fine-grained skeleton part to explore their spatial-temporal characteristic, prompting the intra-class compactness in skeleton space. For semantic space, we propose the learnable semantic-part prompt to improve the inter-class separability of side information further. Afterward, several losses instruct the model to establish the one-to-one and one-to-many correlations between skeleton parts and side information for alignment at the fine-grained level.

The main contributions can be summarized as follows:

- We decompose the skeleton based on human topological structure and introduce part descriptions by LLMs as the side information. In this way, the skeleton and semantic spaces can be effectively aligned at the fine-grained level.
- We propose the two types of prompts, named visual-attribute prompt and semantic-part prompt, to improve the intra-class compactness and inter-class separability among action categories to recognize high-similarity actions.
- Extensive experiments demonstrate the superior performance of the proposed method in ZSL and GZSL settings on NTU RGB+D, NTU RGB+D 120, and PKU-MMD datasets.

## 2 RELATED WORK

### 2.1 Attention-based Zero-Shot Learning

Attention-based ZSL methods aim to augment the critical information of the input, which is effective for recognizing fine-grained categories [21]. Specifically, these methods select important visual regions or sub-attributes by manually calculating the weight matrix and multiplying them on visual [14, 30, 31, 36] or attribute features [11, 17]. However, the abovementioned attention mechanism calculates the element's relation among fixed window sizes, which are deficient in capturing long-term dependency among elements. Inspired by the success of the self-attention mechanism in Transformer [27] in the computer vision community, numerous studies [3, 4, 16, 20] gradually focus on designing various cross-attention mechanisms between their visual and attribute features to explore their correlations in long-term ranges, reducing the visual-semantic gaps. Unlike the above delve into image classification, our work is the first to utilize an attention mechanism to explore the relationship between the skeleton representations and learnable visual-attribute prompts, prompting intra-class compactness for better alignment and recognition.

### 2.2 Skeleton-based Zero-Shot Action Recognition

Existing skeleton-based action recognition methods have poor generality in recognizing unknown actions, prompting researchers to solve this challenge by zero-shot learning [9, 12, 13, 35]. Their common practice is extracting visual features from skeleton sequences with ST-GCN [32] or Shift-GCN [6] and obtaining semantic embeddings from category names or action descriptions with Word2Vec [19] or Sentence-Bert [23]. After that, they design several modules to establish the relationship between the skeleton and semantic spaces at the global level. RelationNet [12] is the first work to learn a deep non-linear metric for matching global visual-sematic pairs. After that, SynSE [9] designed a generative multimodal alignment module based on VAEs to align the global visual features and verb-noun embeddings. MSF-GZSSAR [13] builds on it to extend category names with LLMs for rich information. However, these methods leverage the probability distribution over known categories to recognize actions in the GZSL is unreasonable. Meanwhile, they ignore the spatial-temporal characteristic of actions. SMIE [35] introduces the temporal constraint to align the two modalities globally by maximizing mutual information. Unfortunately, it cannot extend to GZSL. In summary, our method differs in three aspects compared to the abovementioned methods: (1) we introduce the side information and align different modalities at the local fine-grained level rather than the global aspect; (2) we introduce two prompts to improve the intra-class compactness in skeleton space and inter-class separability in semantic space to recognize high-similarity actions; (3) our method can extend to GZSL easily with superior performance by the calibrated stacking method [2].

## 3 METHOD

### 3.1 Problem Definition

Assume we have the skeleton dataset $\mathcal{D} = \mathcal{D}_{tr}^s \cup \mathcal{D}_{te}^s \cup \mathcal{D}_{te}^u$ with $|\mathcal{A}|$ action category names, where $\mathcal{A} = \mathcal{A}^s \cup \mathcal{A}^u$ and $\mathcal{A}^s \cap \mathcal{A}^u = \varnothing$.

**Figure 2: The architecture of the proposed STAR model. In the skeleton stream, we utilize the GCN backbone to extract skeleton representations and then decompose them into several parts based on topology-based partition strategies. The attention-based mechanism and the visual-attribute prompt are devised to improve the intra-class compactness in skeleton space by fully exploring and capturing spatial-temporal characters of the actions. In the semantic stream, we generate the part descriptions of the action as the side information to supply extra fine-grained knowledge. After that, we propose the semantic-part prompt to improve the inter-class separability of these side information with the constraint of the action category name. Finally, we align the multi-part skeleton representations and the corresponding semantic embeddings with the guidance of several losses.**

For the train-set of known categories $\mathcal{D}_{tr}^s = \{x_i^s, y_i^s\}_{i=1}^{N_{tr}}$, $x_i^s$ is the skeleton sequence of known categories with the corresponding category name $y_i^s \in \mathcal{A}^s$. For the test-set of unknown categories $\mathcal{D}_{te}^u = \{x_i^u, y_i^u\}_{i=1}^{N_{te}}$, $x_i^u$ is the skeleton sequence of unknown categories with the corresponding category name $y_i^u \in \mathcal{A}^u$. The goal of the ZSL setting is to construct a model on $\mathcal{D}_{tr}^s$ and then predict the category name on $\mathcal{D}_{te}^u$. For the GZSL setting, we need to predict the category name on $\mathcal{D}_{te}^s \cup \mathcal{D}_{te}^u$ based on the above model, where $\mathcal{D}_{te}^s$ is the test-set of known categories. Unlike the image input, each skeleton sequence $x_i \in \mathbb{R}^{3 \times T \times V \times M}$, where $T$ denotes the sequence frames, $V$ denotes the human joints, $M$ denotes the person number, and 3 denotes the 3D coordinates. For simplicity, we omit the subscript for known (s) and unknown (u) categories during the following training procedures. The framework of our proposed method is illustrated in Fig. 2.

### 3.2 Fine-grained Formulation

Unlike the previous works [9, 12, 13, 35], our method aims to align the skeleton and semantic space at the fine-grained level rather than the global level. From this motivation, we decompose the skeleton sequence into multi-part sequences based on human spatial topological structure. Simultaneously, we generate one-to-one motion semantic descriptions of skeleton part sequences as the side information. This way, we can explore the correlation between the skeleton elements and corresponding side information to transfer knowledge from known to unknown categories.

**(1) Topology-based Multi-part Skeleton Generation.** According to the knowledge of human topology structure, we decompose the skeleton into several joint groups with different fine-grained levels, keeping the same with [29]. Specifically, a two-part strategy means dividing the skeleton into the upper body

and lower body, and four-part and six-part strategies gradually decompose the upper body and lower body into finer partitions, respectively. Afterward, we obtain the fine-grained skeleton representation $f_v = \{f_v^e \in \mathbb{R}^{\widehat{T} \times \widehat{V} \times C}\}_{e=1}^K$ with the spatial topology and temporal continuity by the visual feature extractor $\phi(\cdot)$, as opposed to the previous studies pooling the spatial-temporal dimensions $\widehat{T} \times \widehat{V}$ for global feature $f_v \in \mathbb{R}^C$. The $\widehat{T}$ is the down-sampled temporal dimension, $\widehat{V}$ is the joint group of the $e$-th part, and $K$ is the number of the skeleton parts ($K = \{2, 4, 6\}$).

**(2) One-to-One Side Information Generation.** Compared to the previous studies that use the action category names or their enriched descriptions as the global semantics, we generate the one-to-one spatial-temporal descriptions of the skeleton elements as the extra side information at the fine-grained level. Specifically, we utilize the GPT-3.5 [1] as the expert knowledge base to generate side information by providing appropriate questions. For example, we obtain the side information of head description in action "drink water" with the question "please describe the [head] actions simply when people [drink water]": head tilts back slightly. This way, we can depend on the abovementioned skeleton partition strategies and action category names to generate all actions' fine-grained multi-part skeleton side information. Afterward, we utilize the pre-trained language model as the semantic feature extractor $\psi(\cdot)$ to extract the embeddings of the category name and side information, which can be described as $f_{cn}$ and $f_{si} = \{f_{si}^e\}_{e=1}^K$.

### 3.3 Dual-Prompt Cross-Modality Alignment

The core of the ZSL is to build a bridge by aligning the cross-modality spaces, thereby transferring knowledge between the known and unknown categories. For this, we devise the skeleton representation and semantic embedding network streams to learn a shared

latent space. Simultaneously, we propose the visual-attribute and semantic-part prompts in respective network streams to address the homogeneity challenges among the high-similarity actions.

**(1) Skeleton Representation Network Stream.** The skeleton sequence comprises the human pose coordinates, lacking detailed descriptions of the surrounding environment and human appearance compared to RGB-D videos. Thus, several action categories are highly similar if we only observe the skeleton sequence. With these realities, it is challenging to distinguish highly similar actions such as "reading" and "writing" because only the hand movements are slightly different. Therefore, we employ a cross-attention mechanism with the proposed visual-attribute prompt to improve the intra-class compactness in skeleton space by fully exploring the spatial representations and capturing the long-term dependencies of the skeleton actions.

Specifically, our skeleton representation network stream consists of $K$ branches corresponding to $K$ human parst. Each branch (Attention-based Mechanism) comprises a multi-head cross-attention layer, a feed-forward network (FFN), a learnable visual-attribute prompt, and a projecting matrix. The FFN consists of two linear layers with the Relu activation. The learnable visual-attribute prompt $P_{va} = \{P_{va}^e \in \mathbb{R}^{m \times d_{va}}\}_{e=1}^K$ represents the spatial-temporal motion attribute of the skeleton sequence. $m$ is the hyperparameter of motion attributes, and $d_{va}$ denotes the dimension. The multi-head cross-attention layer uses the $f_v^e$ as the keys $K_v$ and values $V_v$. Meanwhile, it employs the $e$-th part visual-attribute prompt $P_{va}^e$ as queries $Q_v$, which can effectively pay attention to the motions most relevant to each attribute in a given skeleton part sequence. The cross-attention with the visual-attribute prompt in our method can be described as follows:

$$Att_h^v = softmax(\frac{Q_v K_v^T}{\sqrt{d}})V_v, \quad (1)$$

$$\widetilde{f_v^e} = concat(Att_1^v, \cdots, Att_n^v)W_o, \quad (2)$$

where $Q_v = P_{va}^e W_q$, $K_v = f_v^e W_k$, $V_v = f_v^e V_v$, $W_q, W_k, W_v, W_o$ are the learnable weights, $h = 1, \cdots, n$ is the head index, $d$ is the scale factor, and $concat(\cdot)$ is the concatenate operation. After that, we utilize the FFN further to augment the attention-motion features of the skeleton part sequence. To align with the corresponding side information, we further project the output of the FFN $\varphi(\cdot)$ into the shared latent space. It is defined as follows:

$$F_v^e = P_{va}^e W \varphi(\widetilde{f_v^e}), \quad (3)$$

where $W$ is a learnable projecting matrix. To this end, we sum the $F_v^e$ of each skeleton part as the global skeleton representation $F_v$.

**(2) Semantic Embedding Network Stream.** Human actions may have similar movements in several body parts, resulting in the same specific side information. For example, "brush teeth" and "cheer up" have the same side information of head movements: head tilts forward slightly. These identical side information belonging to different action categories can make aligning nonhomogeneous skeleton sequences in the right direction challenging. In other words, these identical side information belonging to various action categories are inseparable in semantic space. To solve this problem, we introduce learnable semantic-part prompts to supplement the category-specific knowledge for side information.

Specifically, we first utilize the CLIP [22] text encoder to extract the category name embeddings $f_{cn}$ and multi-part fine-grained side information embeddings $f_{si} = \{f_{si}^e\}_{e=1}^K$. Then, we add the learnable semantic-part prompt $P_{sp} = \{P_{sp}^e \in \mathbb{R}^{|\mathcal{A}| \times d_{sp}}\}_{e=1}^K$ to side information in the $e$-th part for supplementing and learning the category-specific knowledge as follows:

$$\widehat{f_{si}^e} = f_{si}^e + P_{sp}^e, \quad (4)$$

where $\widehat{f_{si}^e}$ is the augmented side information with category-specific knowledge and $d_{sp}$ denotes the dimension of prompt. Afterward, we utilize two projectors consisting of two linear layers with Relu activation to project the augmented semantic embeddings into the shared latent space for alignment with skeleton space, denoted as the $F_{si}^e$ and $F_{cn}$. In this setting, the category name is the category semantic center that pulls the corresponding augmented side information closer and pushes away non-homogeneous augmented side information as described in Section 3.4, prompting the inter-class separability of the side information.

## 3.4 Model Optimization

To achieve effective optimization, we design multi-part cross-entropy loss, semantic cross-entropy loss, and global cross-entropy loss to guide the training process.

**(1) Multi-Part Cross-Entropy Loss.** To align the skeleton space with the semantic space at the fine-grained level, the initial step is to bring the pairs of part skeleton-semantic closer together. Therefore, we achieve this by calculating the dot product between the skeleton part representation $F_v^e$ and the fine-grained side information embeddings $F_{si}^e$. The multi-part cross-entropy loss can be defined as follows:

$$\mathcal{L}_{MPCE} = -\frac{1}{B \times K} \sum_{i=1}^B \sum_{e=1}^K log(\frac{exp(F_v^{i,e} \times F_{si}^e)}{\sum_{a \in \mathcal{A}^s} exp(F_v^{i,e} \times F_{si}^{e_a})}), \quad (5)$$

where $B$ is the batch size.

**(2) Semantic Cross-Entropy Loss.** To ensure the knowledge that the semantic-part prompt learned is the corresponding category, we suggest promoting the embedding of the augmented side information to have the highest compatibility with its corresponding category name embedding. By doing this, the augmented side information is near its category semantic center, prompting the inter-class separability with each other. The $L_{SCE}$ is defined as:

$$\mathcal{L}_{SCE} = -\frac{1}{K} \sum_{e=1}^K log(\frac{exp(F_{si}^e \times F_{cn})}{\sum_{a \in \mathcal{A}} exp(F_{si}^e \times F_{cn}^a)}). \quad (6)$$

**(3) Global Cross-Entropy Loss.** Besides the alignment at the fine-grained level, we propose the $L_{GCE}$ to align the skeleton and semantic space at the global level. The optimization operation can be described as follows:

$$\mathcal{L}_{GCE} = -\frac{1}{B} \sum_{i=1}^B log(\frac{exp(F_v^i \times F_{cn})}{\sum_{a \in \mathcal{A}^s} exp(F_v^i \times F_{cn}^a)}). \quad (7)$$

We ultimately formulate the overall training loss as shown below:

$$\mathcal{L}_{total} = \mathcal{L}_{MPCE} + \alpha \mathcal{L}_{SCE} + \beta \mathcal{L}_{GCE}. \quad (8)$$

$\alpha$ and $\beta$ are the trade-off parameters, which are set to 0.1.

## 3.5 ZSL/GZSL Prediction

During the inference stage, for a given sample, we can obtain its skeleton global representation $F_v$ and embeddings $\{F_{cn}^a\}_{a=1}^{|\mathcal{A}|}$ of all category names. Then, we utilize the calibrated stacking method [2] to predict the category of the sample, which is defined as follows:

$$\mathcal{A}^* = \arg\max_{a \in \mathcal{A}^u/\mathcal{A}} F_v \times F_{cn}^a - \gamma \mathbb{I}[a \in \mathcal{A}^s], \qquad (9)$$

where $\mathcal{A}^u/\mathcal{A}$ corresponds to the ZSL/GZSL setting, respectively, $\gamma$ is a calibration factor.

## 4 EXPERIMENTS

### 4.1 Datasets

**(1) NTU RGB+D 60 Dataset** [26]. It contains 56880 action samples with multi-modalities, including RGB, depth map, skeleton, and infrared (IR) video, performed by 40 subjects and classified into 60 action categories. For skeleton modality, each sample consists of a maximum of two people, each comprising 3D coordinates of 25 joints. Two official benchmarks are applied: (1) Cross-subject (Xsub): the training set contains 20 subjects, and the remaining subjects are used for testing; (2) Cross-view (Xview): the training set contains view2 and view3, while the view1 make up the test set.

**(2) NTU RGB+D 120 Dataset** [15]. It is the extension of the NTU RGB+D 60, which contains 114,480 action samples of 120 action categories and has the same multi-modalities. Similarly, it also split the dataset into two official benchmarks: (1) Cross-subject (Xsub): the training set contains 53 subjects and the rest of the subjects' data for testing; (2) Cross-setup (Xset): even camera IDs belong to the training and the testing data consists odd IDs.

**(3) PKU-MMD Dataset** [7]. It contains two phases for action recognition with increasing difficulty, which covers the same multi-modalities as the NTU dataset. The actions are collected into 51 action categories, and almost 20000 instances are included. Two official experimental settings are proposed: (1) Cross-subject (Xsub): 57 subjects belong to the training set and 9 subjects for the testing set; (2) Cross-view (Xview): the middle and right views are chosen for the training set, and the left is for the testing set.

### 4.2 Evaluation Protocols

We conduct extensive experiments on the above three datasets in the ZSL and GZSL settings. In the ZSL setting, we compute the Top-1 recognition accuracy of the test samples from unknown categories $\mathcal{D}_{te}^u$, i.e., $Acc$. In the GZSL setting, we calculate the Top-1 recognition accuracy of the test samples from the known categories $\mathcal{D}_{te}^s$ and unknown categories $\mathcal{D}_{te}^u$, denoted as $S$ and $U$, respectively. Then, we also compute their harmonic mean $H = (2 \times S \times U)/(S+U)$, keeping the same as the previous works [9, 12, 13, 35].

### 4.3 Implementation Details

We take the same data processing procedure as the [5]. Following the previous works [9, 12, 13, 35], we employ the Shift-GCN pre-trained on known categories as the visual feature extractor. Meanwhile, we utilize the pre-trained **ViT-L/14@336px** text encoder of CLIP [22] to obtain semantic embeddings. We employ the SGD optimizer to train the model with a batch size of 64 during

**Table 1: The hyper-parameter $\gamma$ and $m$ on NTU RGB+D 60, NTU RGB+D 120, and PKU-MMD datasets.**

| Parameter | NTU RGB+D 60 | NTU RGB+D 120 | PKU-MMD |
|---|---|---|---|
| $\gamma$ | 0.012 | 0.069 | 0.022 |
| $m$ | 100 | 100 | 100 |

the training step. For the NTU series datasets, the initial learning is 0.001 and reduced with 0.1 multiplied at epochs 20 and 30. The weight decay is set to 5e-4. The cross-attention layer is set to 1 with eight attention heads. The Table. 1 shows we set the hyper-parameter $\gamma$ and $m$. We conduct the following experiments on the PyTorch framework with an NVIDIA A100 GPU.

### 4.4 Baseline Settings

We found that previous studies [9, 13, 35] had used the skeleton features and experimental settings provided by SynSE [9] for method validation. However, these settings do not correspond with the official requirements of NTU series datasets, needing more evaluation of their methods' robustness with the subject variation and view changes. At the same time, the number of skeleton features provided by SynSE [9] can not match the official dataset. Therefore, we conduct the experiment based on the official experimental settings (cross-subject, cross-view, and cross-set) of the above datasets. As for known and unknown categories partition strategies that the official requirements lacked, for convenience, we follow the previous studies [9, 35]. For NTU 60, we utilize the 55/5 and 48/12 split strategies, which include 5 and 12 unknown categories. For NTU 120, we employ the 110/10 and 96/24 split strategies. For PKU-MMD II, we take the 46/5 and 39/12 split strategies. To make a fair comparison, we update the performance results of the previous studies under the current experimental settings based on their public code.

### 4.5 Comparison with State-of-the-Art

**(1) Zero-Shot Learning.** In this study, we evaluate the effectiveness of the proposed STAR method compared with other state-of-the-art methods on three datasets in the ZSL setting. As shown in Table 2 and Table 3, our STAR method achieves the best accuracy on NTU series datasets with different unknown-known split strategies. For the cross-subject task, STAR outperforms SMIE by 3.5% and 3.6% in the 55/5 split and 48/12 split of the NTU 60, respectively. Meanwhile, STAR appears to have a similar trend of improvement in NTU 120, indicating that the STAR method can capture the spatial-temporal characteristics of actions at the fine-grained level between cross-subjects. Our STAR method achieves high accuracy for the cross-view task in NTU 60, with 81.6% and 42.5% in different split strategies. Additionally, STAR also achieves state-of-the-art results in the cross-setup task of the NTU 120. These results demonstrate the robustness of our method in dealing with view and setup changes, showing the remarkable capacity to generalize models into unseen views while maintaining accuracy for unknown categories. Lastly, we also explore the different scenarios (PKU-MMD II), as shown in Table 4. We can find that our method dramatically improves accuracy across various tasks and split strategies. Notably,

**Table 2: Comparison of STAR with the state-of-the-art methods on NTU RGB+D 60 dataset in ZSL and GZSL setting.**

| Method | Xsub | | | | | | | | Xview | | | | | | | |
|---|---|---|---|---|---|---|---|---|---|---|---|---|---|---|---|---|
| | 55/5 Split | | | | 48/12 Split | | | | 55/5 Split | | | | 48/12 Split | | | |
| | ZSL | GZSL | | | ZSL | GZSL | | | ZSL | GZSL | | | ZSL | GZSL | | |
| | Acc | S | U | H | Acc | S | U | H | Acc | S | U | H | Acc | S | U | H |
| ReViSE [10] | 69.5 | 40.8 | 50.2 | 45.0 | 24.0 | 21.8 | 14.8 | 17.6 | 54.4 | 25.8 | 29.3 | 27.4 | 17.2 | 34.2 | 16.4 | 22.1 |
| JPoSE [28] | 73.7 | 66.5 | 53.5 | 59.3 | 27.5 | 28.6 | 18.7 | 22.6 | 72.0 | 61.1 | 59.5 | 60.3 | 28.9 | 29.0 | 14.7 | 19.5 |
| CADA-VAE [25] | 76.9 | 56.1 | 56.0 | 56.0 | 32.1 | 50.4 | 25.0 | 33.4 | 75.1 | 65.7 | 56.1 | 60.5 | 32.9 | 49.7 | 25.9 | 34.0 |
| SynSE [9] | 71.9 | 51.3 | 47.4 | 49.2 | 31.3 | 44.1 | 22.9 | 30.1 | 68.0 | 65.5 | 45.6 | 53.8 | 29.9 | 61.3 | 24.6 | 35.1 |
| SMIE [35] | 77.9 | - | - | - | 41.5 | - | - | - | 79.0 | - | - | - | 41.0 | - | - | - |
| **STAR (Ours)** | **81.4** | **69.0** | **69.9** | **69.4** | **45.1** | **62.7** | **37.0** | **46.6** | **81.6** | **71.9** | **70.3** | **71.1** | **42.5** | **66.2** | **37.5** | **47.9** |

**Table 3: Comparison of STAR with the state-of-the-art methods on NTU RGB+D 120 dataset in ZSL and GZSL setting.**

| Method | Xusb | | | | | | | | Xset | | | | | | | |
|---|---|---|---|---|---|---|---|---|---|---|---|---|---|---|---|---|
| | 110/10 Split | | | | 96/24 Split | | | | 110/10 Split | | | | 96/24 Split | | | |
| | ZSL | GZSL | | | ZSL | GZSL | | | ZSL | GZSL | | | ZSL | GZSL | | |
| | Acc | S | U | H | Acc | S | U | H | Acc | S | U | H | Acc | S | U | H |
| ReViSE [10] | 19.8 | 0.6 | 14.5 | 1.1 | 8.5 | 3.4 | 1.5 | 2.1 | 30.2 | 4.0 | 23.7 | 6.8 | 13.5 | 2.6 | 3.4 | 2.9 |
| JPoSE [28] | 57.3 | 53.6 | 11.6 | 19.1 | 38.1 | 41.0 | 3.8 | 6.9 | 52.8 | 23.6 | 4.4 | 7.4 | 38.5 | 79.3 | 2.6 | 4.9 |
| CADA-VAE [25] | 52.5 | 50.2 | 43.9 | 46.8 | 38.7 | 48.3 | 27.5 | 35.1 | 52.5 | 46.0 | 44.5 | 45.2 | 38.7 | 47.6 | 26.8 | 34.3 |
| SynSE [9] | 52.4 | 57.3 | 43.2 | 49.5 | 41.9 | 48.1 | 32.9 | 39.1 | 59.3 | 58.9 | 49.2 | 53.6 | 41.4 | 46.8 | 31.8 | 37.9 |
| SMIE [35] | 61.3 | - | - | - | 42.3 | - | - | - | 57.0 | - | - | - | 42.3 | - | - | - |
| **STAR (Ours)** | **63.3** | **59.9** | **52.7** | **56.1** | **44.3** | **51.2** | **36.9** | **42.9** | **65.3** | **59.3** | **59.5** | **59.4** | **44.1** | **53.7** | **34.1** | **41.7** |

**Table 4: Comparison of STAR with the state-of-the-art methods on the cross-subject task of PKU-MMD II dataset in ZSL and GZSL setting.**

| Method | 46/5 Split | | 39/12 Split | |
|---|---|---|---|---|
| | ZSL | GZSL | ZSL | GZSL |
| ReViSE [10] | 54.2 | 39.1 | 19.3 | 19.0 |
| JPoSE [28] | 57.4 | 52.4 | 27.0 | 37.6 |
| CADA-VAE [25] | 73.9 | 61.7 | 33.7 | 41.1 |
| SynSE [9] | 69.5 | 53.0 | 36.5 | 42.3 |
| SMIE [35] | 72.9 | - | 44.2 | - |
| **STAR (Ours)** | **76.3** | **65.0** | **50.2** | **55.4** |

our proposed method can effectively enhance the recognition of unknown categories by learning the fine-grained relationship between skeleton and semantic spaces rather than global alignment.

**(2) Generalized Zero-Shot Learning.** Here, we compare our proposed STAR with other state-of-the-art methods on three datasets in the GZSL setting. To make a fair comparison with our alignment method, we employ the calibrated stacking method in Section 3.5 as the gating module of all methods to classify known and unknown categories. As shown in Table 2, Table 3, and Table 4, our

proposed STAR achieves the highest accuracy across various tasks and split strategies. Additionally, our method achieves a balanced accuracy regarding known and unknown categories. For instance, our STAR can obtain better performance on known categories and unknown categories as 59.3% and 59.5% for the cross-setup task in NTU 120 with 110/10 split dataset, and thus result in the harmonic mean as 59.4%. In contrast, JPoSE [28] has a considerable margin (19.2%) between their accuracy of known and unknown categories in the same dataset, resulting in poor performance on the harmonic mean. Therefore, this demonstrates that our STAR can learn the discriminant and transferable representations at the fine-grained level, thereby alleviating the problems of domain bias.

### 4.6 Ablation Study

**(1) Influence of Known-Unknown Categories Settings.** Different known-unknown category settings may affect the method's performance. We optimize the experiment procedure and randomly resplit categories into three known-unknown settings to evaluate their robustness. These three split settings are non-overlap, and we compute their average results in the ZSL and GZSL to eliminate the variance, as shown in Table 5. Our proposed STAR performs better than all previous methods in the ZSL, improving by 13.3% and 6.4%, respectively. It demonstrates that STAR is robust in the various known-unknown category settings, and learning prior action

**Table 5: Influence of known-unknown categories settings on NTU RGB+D 60 and PKU-MMD II datasets. The columns of the ZSL and GZSL represent the Top1 accuracy and harmonic mean, respectively.**

| Method | NTU RGB+D 60 55/5 (Xsub) | | PKU-MMD II 46/5 (Xsub) | |
| --- | --- | --- | --- | --- |
| | ZSL | GZSL | ZSL | GZSL |
| ReViSE [10] | 54.7 | 27.4 | 48.7 | 32.8 |
| JPoSE [28] | 56.6 | 44.7 | 39.2 | 31.7 |
| CADA-VAE [25] | 58.0 | 47.1 | 49.0 | 52.7 |
| SynSE [9] | 59.9 | 49.9 | 43.5 | 40.4 |
| SMIE [35] | 64.2 | - | 66.4 | - |
| **STAR (Ours)** | **77.5** | **62.8** | **70.6** | **67.1** |

**Table 6: Influence of topology-based partition strategies at various fine-grained levels on NTU RGB+D 60 and PKU-MMD II datasets. The columns of the ZSL and GZSL represent the Top1 accuracy and harmonic mean, respectively.**

| Strategies | NTU RGB+D 60 55/5 (Xsub) | | PKU-MMD II 46/5 (Xsub) | |
| --- | --- | --- | --- | --- |
| | ZSL | GZSL | ZSL | GZSL |
| Two-part | 77.4 | 67.5 | 73.9 | 63.5 |
| Four-part | 78.1 | 68.6 | 74.7 | 64.5 |
| Six-part | **81.4** | **69.4** | **76.3** | **65.0** |

knowledge from body parts at the fine-grained level can increase generalization performance to recognize unknown actions. In the GZSL, the STAR outperforms the second by 12.9% and 14.4% (relatively 25.6% and 27.3% ), showing that the proposed STAR alleviates the issues of domain bias effectively.

**(2) Influence of Fine-grained Levels.** Under the baseline settings, we explore how different topology-based partition strategies at various fine-grained levels affect the proposed STAR. The strategies we tested include two-part (upper body, lower body), four-part (head, hand-arm, hip, leg-foot), and six-part (head, hand, arm, hip, leg, foot). According to the results presented in Table 6, the six-part partition strategy performs the best on two datasets. This suggests aligning the skeleton and semantic space at the fine-grained level is necessary. Meanwhile, the higher the fine-grained level, the better the recognition performance.

**(3) Influence of Components.** In Table 7, we assess the effectiveness of modules, prompts, and loss functions in the STAR. We can observe that the STAR performs significantly worse without the attention-based mechanism (AM), resulting in a decrease of nearly 2.7%/4.9% and 7.2%/4.5% on two datasets. This indicates that the AM can capture skeleton actions' spatial characteristics and long-term temporal dependency. Additionally, dual-prompts (VPP and SPP) are crucial in improving intra-class compactness in skeleton space and prompting inter-class separability in semantic space. If we remove them, approximately 3.0%/8.5% and 1.9%/5.8% drop on

**Table 7: Influence of different components on NTU RGB+D 60 and PKU-MMD II datasets. The columns of the ZSL and GZSL represent the Top1 accuracy and harmonic mean, respectively. "AM" is the attention-based mechanism, "VAP" denotes the visual-attribute prompt, and "SPP" means semantic-part prompt.**

| Method | NTU RGB+D 60 55/5 (Xsub) | | PKU-MMD II 46/5 (Xsub) | |
| --- | --- | --- | --- | --- |
| | ZSL | GZSL | ZSL | GZSL |
| STAR w/o AM | 78.7 | 64.5 | 69.1 | 60.5 |
| STAR w/o VAP | 78.4 | 60.9 | 67.9 | 58.2 |
| STAR w/o SPP | 79.5 | 63.6 | 69.5 | 61.6 |
| STAR w/o $\mathcal{L}_{MPCE}$ | 78.2 | 60.5 | 58.6 | 57.1 |
| STAR w/o $\mathcal{L}_{SCE}$ | 76.3 | 51.5 | 52.6 | 56.1 |
| STAR w/o $\mathcal{L}_{GCE}$ | 78.6 | 45.8 | 56.2 | 50.9 |
| **STAR (full)** | **81.4** | **69.4** | **76.3** | **65.0** |

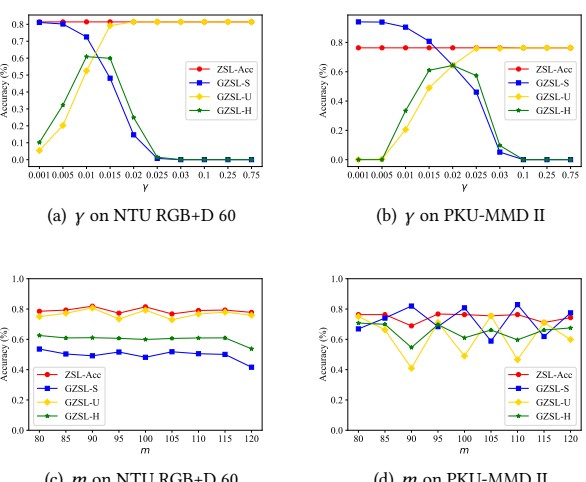

(a) $\gamma$ on NTU RGB+D 60     (b) $\gamma$ on PKU-MMD II

(c) $m$ on NTU RGB+D 60     (d) $m$ on PKU-MMD II

**Figure 3: The influence of hyper-parameters on the NTU RGB+D 60 and the PKU-MMD II datasets.**

the NTU RGB+D 60 dataset, respectively. Moreover, the multi-part cross-entropy mechanism can effectively align the skeleton space and semantic space at a fine-grained level to transfer knowledge, resulting in improvements of 3.2%/8.9% and 17.7%/7.9% on two datasets, respectively. The semantic cross-entropy and global cross-entropy constraints ensure the optimization direction is towards the action category name as the center, effectively prompting the harmonic mean in the GZSL and mitigating the bias problem.

**(4) Influence of Hyper-parameters.** We investigate the influence of the $\gamma$ and $m$ in STAR by selecting a range of values. Our findings, illustrated in Fig. 3(a) - Fig. 3(b), showed that increasing the calibration factor value leads to a decrease in the accuracy of known categories and an increase in the accuracy of unknown categories, enabling us to determine the optimal calibration factor

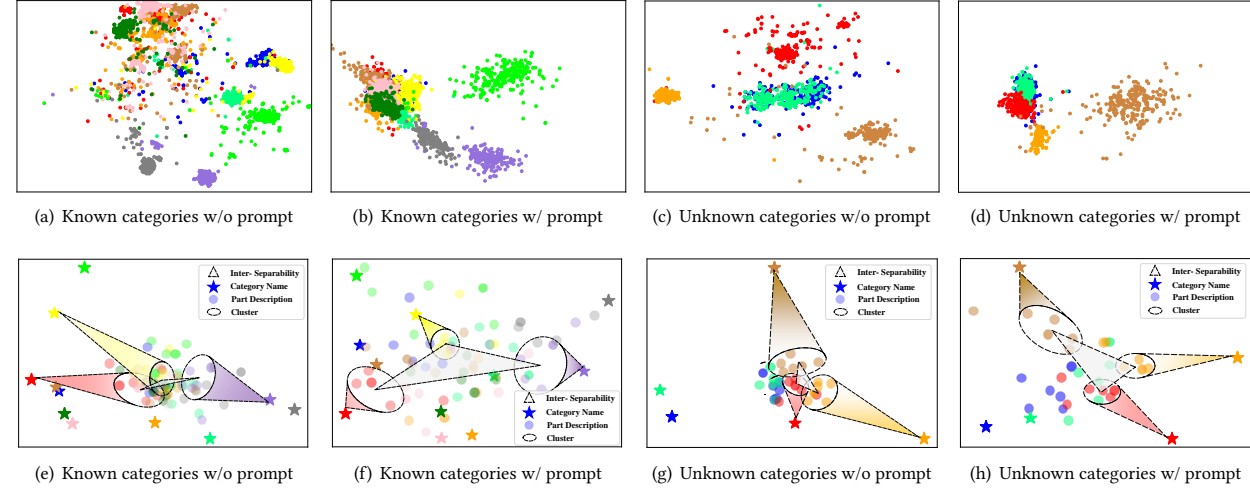

Figure 4: t-SNE visualizations of skeleton and semantic spaces for known and unknown categories. The color denotes different known/unknow categories random selected from the cross-subject task of NTU RGB+D 60 dataset under the 55/5 split settings. The first row ((a) - (d)) represents the skeleton space, and the second row ((e) - (h)) represents the semantic space.

value for GZSL. This calibration factor effectively balances the accuracy of known and unknown categories, achieving the trade-off between them to alleviate the bias problem. Additionally, our experiments with motion attribute $m$ in Fig. 3(c) - Fig. 3(d) show that the recognition accuracy remains stable with the number of attributes increases, indicating the robustness of proposed prompts.

## 4.7 Qualitative Analysis

**(1) t-SNE Visualizations of Spaces.** As shown in Fig. 4, we plot the t-SNE visualization of the skeleton and semantic spaces for known and unknown categories on the NTU RGB+D 60 dataset. Fig. 4(a) - Fig. 4(d) represent the distribution of skeleton features with the visual-attribute prompt or not, showing the capacity to improve the intra-class compactness of skeleton space over the known and unknown categories. Meanwhile, we find that the area of the triangle in Fig. 4(e) - Fig. 4(h) increases with the addition of the semantic-part prompt, demonstrating that the semantic-part prompt can improve the inter-class separability of semantic space. Besides, the optimized part descriptions are pulled closer to their semantic center (category name) for better alignment with the decomposed human skeleton.

**(2) Visualization of the Confusion Matrices.** Here, we compare the capacity of STAR with SMIE [35] for distinguishing high-similar skeleton actions that never appeared before. We draw the confusion matrices of two methods for unknown categories on the cross-subject task of the NTU RGB+D 60 dataset under the 55/5 split setting, as shown in Fig. 5. It shows that SMIE is deficient in recognizing skeleton actions with subtle differences, such as writing and reading. These skeleton actions throughout the body only have differences in the hand because the appearance and environment information are dropped. In contrast, our method STAR can distinguish these highly similar skeleton actions based on the prior

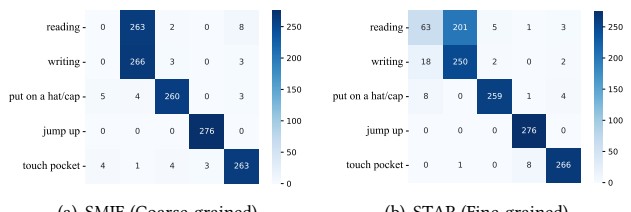

Figure 5: Confusion matrices for unknown categories on the cross-subject task of NTU RGB+D 60 under the 55/5 split setting. (a) represents the confusion matrix of SMIE [35] method. (b) represents the confusion matrix of STAR (our method).

knowledge learned from the known categories, showing the necessity of aligning the skeleton and semantic spaces at the fine-grained level.

## 5 CONCLUSION

In this paper, we propose a novel method via **S**ide information and dual-promp**T**s learning for zero-shot skeleton **A**ction **R**ecognition (STAR). Firstly, our STAR decomposes the skeleton sequence into several topology-based human parts for alignment at the fine-grained level with the introduced side information concerning multi-part descriptions of human body movements. Secondly, the visual-attribute and the sematic-part prompts are designed to improve the inter-class compactness in the skeleton space and intra-class separability in the semantic space. Extensive experiments on three popular datasets demonstrate the superiority of our approach, especially in recognizing highly similar actions.

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
