# OpenReview forum: "Fine-Grained Side Information Guided Dual-Prompts for Zero-Shot Skeleton Action Recognition"
_acmmm.org/ACMMM/2024/Conference — MM2024 Poster_

### Official Review · Reviewer_Nj73 · 2024-04-29

**Rating:** 4
**Confidence:** 3

**Summary:**

This paper introduces a novel fine-grained alignment framework for zero-shot skeleton-based action recognition, which mainly utilizes the human topological structure and part desriptions as side information to capture fine-grained  information to enhance global action representations. To further recognize high-similarity actions, visual-attribute prompt and semantic-part prompt are designed to improve the intra-class compactness and effectively distinguish different actions, respectively. Experimental results demonstrate that this method is superior to previous zero-shot action recognition approaches, and achieves the best performance on three public action datasets.

**Strengths:**

1. A new zero-shot skeleton action recognition method is proposed, which relies on side information and dual-prompt learning to effectively model local parts indormation and enhance global action semantic representation. Different from existing global representation approaches, this method utilzes abundant fine-grained visual-language body features to establish more complete and informative action represention. I think that this is a novel and creative method in zero-shot skeleton action recognition.
2. Experimental results are very sufficient and this paper further provides the GZSL relevant results.
3. Two novel learnable prompts are designed in the visual and textual streams to further improve action classification performance.
4. The paper is well written and very easy to read.

**Limitations:**

1. There is no introduction to traditional skeleton-based action recognition in the related work section, and there is no detailed discussion on the differences between this approach and related work [1].
2. The structure of two learnable prompts and how they respectively learn from visual and textual features has not been described in detail.
3. some questions:
(1) Have the authors used any other methods besides the GPT-3.5 used in the paper to generate body part descriptions.
(2) Why not use GPT to generate detailed text descriptions for action categories as well?
(3) What is the initialization of two prompts like？
(4)What does the attribute in the visual attribute prompt represent?
(5) In addition to t-sne visualization, I am very confused about the usage of a learnable visual attribute prompt to capture the temporal and spatial information of each body part. Why is it very helpful to reduce intra-class variances, and may provide some new experimental results and visulizations.

[1] W. Xiang, C. Li, Y. Zhou, B. Wang and L. Zhang, "Generative Action Description Prompts for Skeleton-based Action Recognition," 2023 IEEE/CVF International Conference on Computer Vision (ICCV), Paris, France, 2023, pp. 10242-10251, doi: 10.1109/ICCV51070.2023.00943. keywords: {Representation learning;Training;Costs;Source coding;Semantics;Thumb;Benchmark testing},

**Suitability:**

3

---

### Official Review · Reviewer_ZTmd · 2024-05-21

**Rating:** 5
**Confidence:** 3

**Summary:**

In this paper, the authors introduce a zero-shot learning framework named STAR for skeleton-based action recognition. The framework utilizes finer-grained detail information for both skeleton and semantic features, enhancing the alignment between visual and semantic spaces.

**Strengths:**

1. The paper is well-written and the work on skeleton-based zero-shot action recognition is comprehensive.
2. The transition from global information to the use of fine-grained information is a reasonable idea. And STAR can be applied to both ZSL and GZSL.
3. In zero-shot tasks, the impact of different seen and unseen splits is significant. The author has taken this into consideration, and the experiments and figures are both extensive.
4. STAR achieves the SOTA performance on three large skeleton datasets and the ablation studies confirm the effectiveness of each module.

**Limitations:**

1. The "related work" section could benefit from including a discussion on skeleton action recognition, providing readers with essential knowledge about skeleton data.
2. Regarding the extraction of skeleton visual features, is Shift-GCN capable of extracting visual features for each specific part?
3. How can the reliability of large language models (LLMs) in providing fine-grained action descriptions be ensured?
4. The author asserts that certain actions share similar head movement information, but it is not clearly explained how learnable semantic-part prompts address this issue.

**Suitability:**

2

---

### Official Review · Reviewer_XjxS · 2024-05-21

**Rating:** 2
**Confidence:** 4

**Summary:**

This paper proposes a novel method for fine-grained skeleton-based zero-shot action recognition (STAR) that leverages side information and dual-prompt learning. Specifically, the method decomposes the skeleton into parts and introduces side information about multi-part body movements for fine-grained alignment, while also designing visual-attribute and semantic-part prompts to improve intra-class compactness and inter-class separability for distinguishing high-similarity actions.

**Strengths:**

The research achieved fine-grained alignment of multiple body parts and designed visual attribute prompts and semantic part prompts to distinguish actions with high similarity. The experiments are comprehensive, including the addition of GZSL experiments and the implementation of two experimental settings.

**Limitations:**

The last paragraph of the Introduction is flawed. It only explains how to solve the first two challenges but fails to clarify how to address the GZSL issue (challenge 3).

The description of visual-attribute prompt and semantic-part prompt in the methodology section is not clear enough. More details should be provided to strengthen the explanation of these prompts. Additionally, the representation and explanation of the formulas are unclear, and they need to be clarified further.

The results achieved on the NTU120 dataset in the experimental section are inferior to those reported in the SMIE article. More experiments are needed to surpass the numerical values provided in the SMIE article.

The PKU-MMD dataset has two experimental settings, but only one is implemented in this paper. The other setting needs to be supplemented.

The DeViSE and RelationNet methods are not included as comparison methods in the experimental comparison. Their inclusion is necessary for a comprehensive evaluation.

**Suitability:**

3

---

### Official Review · Reviewer_fH1z · 2024-05-23

**Rating:** 4
**Confidence:** 2

**Summary:**

The paper proposes a zero-shot skeleton action recognition method, which focuses on fine-grained alignment between skeleton representation and semantic descriptions via side information. Moreover, the method uses dual-prompts learning to improve the intra-class compactness and inter-class separability among action categories. Extensive experiments demonstrate the superior performance of the method.

**Strengths:**

1. The technologies are sound reliable.
2. The experiments are effective.

**Limitations:**

1. The paper is not well written.
2. In Fig.2, some symbols are not described. The figure is confusing.
3. I suggest that the author add some figures to describe the method in detail. E.g., generation of side information, training and testing phases, visual comparison, etc.
4. How to obtain the action category of one sequence.
5. How to ensure the accuracy of the text generated by GPT during training?

**Suitability:**

2

---

### Meta-Review · Area_Chair_qJJ1 · 2024-06-28

**Recommendation:** Accept (Poster)
**Confidence:** 5

**Metareview:**

This paper initially received a mix of reviews. However, after the rebuttal, reviewers upgraded their review and agreed to accept the paper. The AC thoroughly considered the paper, reviews, rebuttal, and subsequent discussions, ultimately deciding to accept it. The authors are encouraged to incorporate the reviewers' suggestions to enhance the final version.